# An integrated methodology to develop a standard for landslide early warning systems

Teuku Faisal Fathani[1], Dwikorita Karnawati[2], Wahyu Wilopo[2]

[1]Department of Civil and Environmental Engineering, Faculty of Engineering, Universitas Gadjah Mada, Yogyakarta 55281, Indonesia
[2]Department of Geological Engineering, Faculty of Engineering, Universitas Gadjah Mada, Yogyakarta 55281, Indonesia

*Correspondence to*: Teuku Faisal Fathani (tfathani@ugm.ac.id)

**Abstract.** Landslides are one of the most widespread and commonly occurring natural hazards. In regions of high vulnerability, these complex hazards can cause significant negative social and economic impacts. Considering the worldwide susceptibility to landslides, it is necessary to establish a standard for early warning systems specific to landslide disaster risk reduction. This standard would provide a guidance in conducting landslide detection, prediction, interpretation, and response. This paper proposes a new standard consisting of seven sub-systems for landslide early warning. These include risk assessment and mapping, dissemination and communication, establishment of disaster preparedness and response team, development of evacuation map, standardized operating procedures, installation of monitoring and warning services, and building a local commitment to the operation and maintenance of the entire program. This paper details the global standard with an example of its application from Central Java, one of 20 landslide prone provinces in Indonesia that have used this standard since 2012.

## 1 Introduction

Landslides are one of the most widespread and frequent natural and anthropogenic hazards. Landslide mitigation is conventionally associated with the physical triggers, such as precipitation, earthquakes, and slope interference, among others (Ramesh, 2014; Wieczorek and Glade, 2005; Senneset, 2001). There are multiple approaches to reducing landslide risk that can be broadly classified as structural or non-structural. An example of non-structural disaster risk mitigation is to increase the preparedness of the community through the implementation of an effective and reliable early warning system (Bednarczyk, 2014; Michoud et al., 2013).

Early warning is a timely provision of information through appropriate institutions that enable exposed individuals to take precautionary actions (UNEP, 2012). There are many definitions of early warning systems (Medina-Cetina and Nadim, 2008), but a common reference from UN-ISDR (2006) states that a comprehensive and effective people-centered early warning system consists of four interrelated key elements, namely risk knowledge, monitoring and warning device, dissemination and communication, and response capability. Together and through refinement these elements can evolve into the development of an effective landslide early warning system (Intrieri et al., 2012).

Therefore the concept and application of landslide early warning systems is not new. Thiebes (2011) and Thiebes and Glade (2016) outline approaches implemented in different parts of the world. The majority of these systems included the implementation of various technological and modeling methods to predict landslide. For example, a system installed in the Citarum River Catchment in Indonesia uses hydrological-geological modeling to predict the landslide (Apip et al., 2010). However, cultural, economic, social, and demographic considerations are often left out of the design compared to the other technical aspects in the currently developed early warning systems (Thiebes and Glade, 2016). Furthermore, training on early warning systems, and in particular the proper precautionary responses should be followed up not only by researchers but also by the practitioners on the national and local level (Fathani and Karnawati, 2013; Fathani et al., 2014). Therefore,

there is a necessity to create a universal standard for landslide early warning systems that puts more specific emphasis on the role of the community and social aspects in general.

A global standard would also support countries in the implementation of revised international frameworks for managing risk such as the Sendai Framework for Disaster Risk Reduction (SFDRR) 2015-2030. The framework declared that national and community resilience against disaster is obtained through disaster prevention and risk reduction. Two components highlighted for the delivery of the 'priorities for action' are risk assessment and early warning in order to respond effectively to a disaster. Specifically, by implementing a simple, low-cost early warning system and improving the dissemination of information at local and national levels. The need for a legal standard is important to exemplify the early warning capacity and increase the community compliance (Eidsvig et al., 2014). Considering that landslide disasters commonly occur at a local area having similar geomorphology and geological condition, the proposed standard within this paper only addresses local scale landslide. This standard is aimed at empowering individuals and community at risk to act in sufficient time to reduce the number of the casualties (UN-ISDR, 2006). This standard was developed by considering the capacities of local communities through a socio-economic, cultural and educational lens in order for the standard to be locally understandable and owned.

## 2 Methodology to Develop a Universal Standard

As the types and mechanisms of landslide early warning systems vary, a universal standard should be developed so that uniformity in the implementation of an early warning system and improvement of community and stakeholders preparedness in landslide-prone areas can be obtained (WMO, 2010). This system is intended to be implemented by the local authorities, universities, research centers or the private sector at many landslide prone areas around the world that have different physical conditions and a unique socio-economic-cultural characteristics. In order to ensure the widespread applicability of this system, a standard should provide consistent and clear technical terms and definitions, as well as requirements and general phases in the implementation of landslide early warning systems. It should also regulate the standardization of commonly used monitoring equipment, warning criteria, the color of the lights and the sound of the sirens, the style of evacuation maps, and types of disaster preparedness and response organization. Although flexibility is required during the implementation, it should adhere to the key elements contained within the standard.

The proposed standard adopts a hybrid socio-technical approach in disaster risk reduction (Karnawati et al., 2011, 2013b) for each single and local-scale landslide. This approach needs inter-disciplinary roles to support disaster risk reduction in the context of community development. The technical approach plays a role in the risk assessment and installation of hazard monitoring and warning services. However, based on experience in installation of landslide early warning instruments in Southeast Asian countries since 2008, focusing only on the technical approach does not guarantee the effectiveness and sustainability of the systems (Fathani et al., 2014). In order to overcome this problem, applying a social lens plays a key role in the success of the program, particularly in terms of establishing the disaster preparedness protocol, developing the response team, evacuation map, a standard operating procedure, and enhancing local commitment. Both approaches are supported with continuing education and research. However, it should be noted that this hybrid technique should be low cost using simple methods and technology so that the community can understand, operate, and maintain it properly (Karnawati et al., 2013a).

Taking into account the four key elements of a people-centered early warning system (UN-ISDR, 2006) and the hybrid socio-technical approach for disaster risk reduction (Karnawati et al., 2013a, 2013b), a universal standard for landslide early warning systems which comprises seven sub-systems is proposed as elaborated in Fig. 1. It can be clarified that monitoring and warning services that to date are considered as the core of early warning systems will remain an important part of the

disaster management program. In Asia, people depend on hilly areas not only as their dwelling place, but also for agriculture and livestock farming (Arambepola and Basnayake, 2014). As landslide affected areas are commonly isolated, the implementation of the early warning system with seven sub-systems is expected to increase the capacity of the locals as first responders and eventually support the establishment of resilient villages/districts that will contribute to national resilience (Fathani et al., 2014; Karnawati et al., 2013b). The following sections will explore each of 7 elements within the proposed standard of landslides early warning systems.

## 2.1 Risk assessment and mapping

Risk assessment and mapping is carried out by technical, institutional, and socio-economic-cultural surveys within the vulnerable community. The survey is conducted by the local authority along with the local community and supported by researchers and experts. This assessment is an important first step to determine the strategy for the implementation of the system from various aspects. This systematic approach serves to identify the hazardous and safe zones and to prioritize the location of hazard monitoring and warning devices installation (Michoud et al., 2013).

*The technical survey* is performed to understand the geological conditions in certain areas, especially to determine landslide susceptibility and stable zones (Collins, 2008). This survey is also conducted to gather information on the history of landslide movement, damaged infrastructures, and signs regarding mass movement such as crack, subsidence, appearance of spring water, fracture on structures, and tilting of poles and/or trees. During technical surveys, information on lithology and distribution of soil and rock formations should also be included. By examining the results of technical surveys, the authority and community could identify the potential instability of slopes, predict the impact, and determine the placement of the landslide monitoring and early warning instruments.

*The institutional survey* is performed in order to understand whether an established institution or a local organization exists to monitor and mitigate landslide hazards.

*The cultural-economic survey* is conducted to gather information on community demographics, such as population (household), age, education, financial situation (vehicle and livestock ownership), and culture, to identify socially acceptable entry points for the joint implementation of an early warning system. Information on potential vulnerable inhabitants and infrastructure due to landslide is important to determine the risk level in a certain area.

*The social survey* is performed to understand the community's understanding of landslide hazards and address the social issues and gaps within the community. The community's eagerness and motivation to actively participate is prerequisite to regulate strategy of risk reduction programs that are suitable with the local socio-economic-cultural conditions. To increase people's awareness, one of the empowerment programs is training and continuing education. This activity will give knowledge and increase people's capacity to be able to decide what needs to be done in order to prevent and protect themselves from landslides.

## 2.2 Dissemination and communication

Dissemination of information relating to landslide disasters aims to provide comprehension and understanding to the community, and to understand the community's aspirations (Jaiswal and van Westen, 2013). The dissemination and communication process is equally important to assess the community risk perception and its efficiency (Eidsvig et al., 2014; Lateltin et al., 2005). Methods and materials for dissemination are tailored based on the preliminary data of the risk assessment and mapping that have been performed. This includes the definition, mechanisms of occurrence, controlling and triggering factors, the symptoms, and the mitigation options of landslide which includes early warning devices, warning levels and warning signs. The inclusion of risk knowledge is also important during the dissemination process (IEWP, 2008).

The aim of dissemination is that the people have a better understanding of the landslide characteristics threatening their area (e.g. causes and mechanisms) and how to minimize risk. Furthermore, during the dissemination and communication process it may be possible to identify the key people who have a strong commitment as forerunners in the establishment of the disaster preparedness team.

**2.3 Establishment of disaster preparedness and response team**

A disaster preparedness and response team is established through community consultation facilitated by the local authority or related agencies. The disaster preparedness and response team mentioned in this section is similar with a "disaster prevention volunteer" in Chen and Wu (2015). The appointment of this team is based on the ability of each member in landslide preparedness, prevention, mitigation, emergency response and post-disaster management. The team should consist of at least
10 a chairperson, data and information division, refugee mobilization division, first aid division, logistics division, and security division. Other divisions included in the team may be added according to the needs of the community and must remain in accordance with the purpose of an early warning system. Each division consists of at least three people or in proportion to the number of population. In addition, it should be composed of permanent residents who live in the hazard prone area.

The disaster preparedness team is tasked to conduct all preparedness and response activities, including mobilizing the
15 community to support the technical system effectively. The team is in charge of determining landslide risk zones and evacuation routes that are verified by the local authority or experts and mobilizing people to evacuate before the landslide occurs. All members of the disaster preparedness team are required to participate in an "orientation and training program" then finally selected to represent a particular division (Arambepola and Basnayake, 2014). The team is then responsible to disseminate all information mentioned in the evacuation map and to train the local community regularly to increase their
awareness on how to implement the standard operating procedures for evacuation. This process of continuing education is essential because even in a community exposed to landslide risk, many of them are not aware of the risks they have (Calvello et al., 2016). It is emphasized that the community actively participate, because one of the indicators of preparedness of the community will be their own activism that will have a direct impact on the mortality rate after the disaster (Chen and Wu, 2014). In addition, the team is also responsible for operating and maintaining the installed monitoring devices and
conducting a regular evacuation drill at least once a year.

**2.4 Development of evacuation map**

An evacuation map that provides information on the unsafe zones and areas safe from landslide hazard should also include secure evacuation routes, and strategic gathering locations (assembly points). The landslide risk zones and evacuation routes serve as operational guidelines for the disaster preparedness and response team and the vulnerable community to gather in an
30 assembly point and subsequently to evacuate by following a predetermined route. The map is developed by the disaster preparedness and response team after having a basic training on hazard mapping. The locals are invited to contribute by adding new landslide features that were found during field inspections. The minimum information provided by an evacuation map, are (Karnawati et al., 2013c):
a. High-risk and low-risk (safe) zones;
b. Landslide features: crown, crack, movement direction, landslide boundary and spring;
c. Houses and important facilities: school, mosque, church, community health center, offices, and landmarks;
d. Alert post, assembly point(s) and evacuation shelter(s).
e. Installation point of early warning system;
f. Streets and alleys;
40 g. Evacuation route(s).

The evacuation map is very simple and easily understood by the local people, even for those who have a low level of education. In this case, the applied village hazard map may not comply with all of the technical requirements in mapping but it contains all of the basic information to provide guidance when conducting evacuation (Karnawati et al., 2013c).

## 2.5 Establishment of standard operating procedure

The Standard Operating Procedures (SOP) serves as a guide for the disaster preparedness team and the community living in a hazard prone area, when facing all hazard levels. The numbers of hazard levels are adjusted for the local conditions taking into account physical characteristics, geomorphological conditions, affected area, rate of movement, and accessibility to a safer area. The SOP contains the response procedures for the disaster preparedness team and the community, specific to the alert. The SOP was prepared based on the discussions and agreements of each division under the direction of the local

authority and relevant stakeholders. Table 1 shows typical standard operating procedures for evacuation.

The warning levels are determined based on the monitoring data that are verified by trained officers with visual ground check. At some areas, the local community might decide to have green level as the lowest level, which means no landslide threats. Key activities during green level are regular coordination between the disaster preparedness and response team, as well as regular checks of the monitoring and warning devices. However, at most landslide prone areas, the community

decide to have "caution" as the lowest level. The determination of Level 1 (CAUTION: landslide possible) is usually based on the results of rain gauge measurement. Level 2 (WARNING: landslide likely) and Level 3 (EVACUATE: landslide occurrence imminent) are determined when the rain intensity exceeds the determined warning thresholds, along with the increase in groundwater, and the increase in other landslide indications in terms of ground surface or slip surface deformation. The determination of warning thresholds strongly depends on landslide types, geological condition, and

previous knowledge in order to analyze the long time series. However this manuscript does not deal with the determination of the warning thresholds. The warning thresholds are determined by experts after analyzing the monitoring data in the area or other areas with similar landslide conditions. Further criterion of landslide early warning threshold may be found in Wieckzorek and Glade (2005) and Guzzetti et al. (2008). In this case, the determination of warning thresholds is carried out relative to the Indonesian rainy season but in the other parts of the world, the timing should reflect local conditions.

The SOP in Table 1 clearly stated that in Level 1 the disaster preparedness and response team should conduct community coordination and data collection from the local people. During data collection, the officers should inform the people on the increase in hazard level, appropriate preparation, evacuation route, the location of assembly point, and also ask them to monitor their environment. In Level 2, the information and data division should conduct visual ground check to the monitoring devices, and if the landslide indications had already been verified, they should evacuate the vulnerable group.

Furthermore, in Level 3 all residents are evacuated based on the guidance in the evacuation map.

The role of the local authority in each level is to receive reports from the head of the team, check the location, and provide emergency support to the evacuated residents. The establishment of SOP is important to clearly define the role and responsibilities of the disaster preparedness team and the community when dealing with specific landslide alert (Michoud et al., 2013). However it is important to identify the type of communication system and overall operation procedure that will

work best locally.

## 2.6 Installation of hazard monitoring, warning services and implementation of evacuation drill

Landslide monitoring and warning devices can be in the form of either remote, proximal, or close-range monitoring systems. Barla and Antolini (2016) showcase common conventional monitoring modules that involve the operations connected to the installation, data acquisition and processing of the in-situ geotechnical instrumentation (extensometer, tiltmeter,

inclinometer, piezometers, etc.) and of further remote sensing equipment which can be adapted for landslide monitoring (i.e. terrestrial laser scanner, total stations, photogrammetric techniques, etc.). Based on previous experiences Michoud et al. (2013) stated that simplicity, long-term robustness, presence of multiple sensors, proper maintenance budget, and power and communication lines backups are among the important precursors of an effective and successful monitoring network.

This proposed methodology focuses on the conventional monitoring module, since it is commonly used at the community level to produce a local and immediate warning communication. The conventional monitoring devices consist of the integration of various instruments to measure rainfall (rain gauge), to measure the ground movement (extensometer, tiltmeter, inclinometer, and pipe strain gauge), to measure the fluctuation of groundwater level and pore water pressure (piezometer), and survey stakes with or without a telemetry system (Thiebes and Glade, 2016; Yin et al., 2010). Each

monitoring device sends designated information concerning the hazard level directly to the community and to the local control center. The mechanism of data transmission of inter human-technical sensors is shown in Fig. 2. The trained officer role is to conduct a visual ground check on the monitoring equipment and warning device in order to identify if a false warning has happened (shown in dotted line). On the other hand, the trained officer might identify a clear initiation of landslide movement in the field when the equipment has a technical error to record the symptoms. This system proposes

three paths for issuing the warning: (1) local control center; (2) local authority; (3) real-time interface or SMS blasting. Each device is equipped with lights in different colors and sirens with different sounds to show the hazard levels namely CAUTION-WARNING-EVACUATE. Sirens sound off when the surface/ground movement or rainfall intensity, pore water pressure or groundwater exceeds the critical thresholds. The disaster experts should determine the warning threshold that may trigger potential landslides, by involving the local community. This involvement would eventually increase the

acceptance of false alarms and missed events. The warning and monitoring networks are all equally important and will succeed in its purpose if all the components are installed correctly (Angeli et al., 2000).

    An instrument to measure changes in slope inclination (tiltmeter) is installed in areas susceptible to slope inclination change. Disaster experts should determine the critical limit of soil movement in degree (°) minute$^{-1}$ or hour$^{-1}$, in the X-Y direction (N–S and W–E). If the instrument indicates slope inclination change that exceeds the critical limit, then it triggers the

warning mechanism. The instrument to measure soil crack (extensometer) is installed in areas susceptible to ground movement. This device has critical limits in mm/minutes or mm/hour, depending on the field condition. With the same method, inclinometer, pipe strain gauge, and multi-layer movement devices are installed to detect movement on slip surface. Other devices to detect mass movement can be installed and integrated with this system to give timely and proper warning to the community.

In telemetry-based monitoring, every movement at the ground and slip surface, rainfall intensity-duration and groundwater fluctuation are being recorded by sensors and transmitted to an operations control center. The local server analyzes the data by taking into account the critical limit of ground movement and rainfall intensity-duration. Cautiousness is important in installing the early detection sensor in high-risk zones with a high number of people at risk. Determination of the installation location is based on zonation of landslide risk. The installation should be done together with the locals so that they develop

greater sense of belonging and responsibility towards the devices and an entire system. The devices should be installed appropriately taking into account the geological condition, the existing symptoms, and landslide volume and potentially affected area. To realize a community-based landslide early warning system, the monitoring and early detection devices should use the most effective and adaptive technology (Fathani and Karnawati, 2013). Once the devices are installed, the teams are formed, the evacuation map and SOP are made available, and an evacuation drill is conducted to ensure the

functionality of the devices and the community's responses. This annual drill will embody the "risk consciousness" as mentioned in Jaiswal and van Westen (2013) study. Evacuation drills are carried out based on a scenario drawn up according

to the SOP (Table 1). It serves to train vigilance, preparedness, and responsibility of the disaster preparedness team during the time that the early detection devices indicate potential landslide. In addition, the evacuation drill is also aimed to introduce and familiarize the local community with the sounds of the sirens from each stage of the early detection devices, and to train people on evacuation.

## 2.7 Commitment of the local authority and community

The commitment of the local government and the community is crucial in the operation and maintenance of the system so that all activity stages included in the SOP run well. This commitment is expected to provide constant communication among all related stakeholders to ensure the result of the system (Lacasse and Nadim, 2009). The duty and responsibility in terms of ownership, installation, operation, maintenance, and security of an early warning system are adjusted to the condition in each location and are agreed upon by the authority, the community, and the private sectors. The legal aspects are very important to ensure the implementation of the systems, however it is not discussed in this paper. Based on experiences, sustainability of the system is assured with keen involvement of local government (Kafle and Murshed 2006). More advanced efforts would include the landslide early warning system as an extension to the local government work program. To ensure future improvement on disaster risk reduction, it is also important to conduct periodical analysis and audit on the community engagement and involvement of the relevant authorities (Arambepola and Basnayake, 2014; Hernandez-Moreno and Alcantara-Ayala, 2016).

## 3 The Result of the Implementation of the Proposed Methodology

Since 2008, landslide monitoring systems have been implemented in Indonesia, starting with a manual monitoring device, paper-recorded device, utilization of data logger through to using real-time monitoring systems (Fathani and Karnawati, 2013). Since 2012, the newly proposed standard has been trialed in 50 districts throughout 20 provinces in Indonesia and Myanmar. Locations of the implementation are indicated in the landslide risk map of Indonesia (Fig. 3). According to Indonesian National Disaster Management Authority-BNPB (2011), out of 453 districts in Indonesia, 42 of them are classified as having high potential landslide risk, whereas 228 districts have medium potential landslide risk. In total, 41 million people are exposed to landslide hazard. Therefore, the management of landslide risk is the main priority in the National Plan of Disaster Management (BNPB, 2011).

As an example, the implementation of the proposed methodology in Banjarnegara District, Central Java Province, Indonesia is explained. Risk assessment was conducted by technical, institutional, and socio-economic-cultural surveys of the community performed together with the community. The activity began with the technical surveys to identify physical symptoms of the landslide hazard, such as cracks, depressions, upheavals and spring. The surveys were conducted with several key people whom the local authority and experts directly trained on identifying the early symptoms of landslide, the mechanism of slope movement and its processes, and the preventive measures (Cruden and Varnes, 1996). Further, a socio-economic-cultural survey was also conducted with the local people through in-depth interviews or focus group discussion. The results of the preliminary surveys were then discussed in a meeting to communicate and to disseminate the information and at the same time to establish the disaster preparedness team (Fig. 4). One of the important results from the socio-economic-cultural survey was a comprehensive and accurate community data collection, e.g. number of households, vulnerable groups, amount of vehicles and cattle.

The results of risk assessment were used to determine the location of landslide early detection devices. Usually, the number of devices is limited and not all can be installed in the high risk zone. Due caution and circumspection is essential to decide where the devices should be installed. Generally, the devices were installed at the most critical areas based on the ground

symptoms that show rapid movement compared to other zones. Other factors that determine installation location are the number of exposed lives, accessibility, device security, land ownership, etc. The location for the device installation was identified by inviting active participation of the local community.

In parallel with the installation of monitoring equipment and warning devices, the evacuation map and evacuation SOP were being developed. Examples of an evacuation map developed at a village in Banjarnegara District of Central Java Province are shown in Fig. 5. This demonstrates a simple and straightforward evacuation map which is easy for the local people to understand and follow. The map contains information regarding low to high hazard zones, evacuation route and the location of the installed monitoring devices (rain gauge, extensometer, tiltmeter, inclinometer, and piezometer) and warning devices (sirens). Furthermore, the map also contains important landmarks in the area, and locations of each house with detailed information of the house number and the name of the head of the household. Evacuation routes are shown by arrows and forbidden zones are delineated.

The evacuation SOP is composed in compliance with the newly proposed standard in Table 1. The SOP is divided to three warning levels: CAUTION-WARNING-EVACUATE based on each location's characteristics. In each level, a comprehensive explanation on what needs to be done, who is in charge, how to respond, etc. is provided. In level CAUTION, the main activity is the coordination of disaster preparedness team and data collection. The main activity in level WARNING is evacuation of vulnerable group (sick people, disabled, elderly, children, pregnant women) and officers have to conduct visual check on the devices and landslide hazard zone. The main activity in level EVACUATE is to evacuate all population in the area to the temporary shelter. Disaster preparedness team is also required to monitor the situation and close access to any high threat zones. The final step of this methodology is to carry out an evacuation drill for each warning level. The disaster preparedness team conducts their tasks by referring to the existing SOP and evacuation map. Facilitators from local authority, experts from university/research centers, and NGOs observe the process and ultimately give an evaluation at the end of the drill. Unlike the evacuation systems in tsunami, volcanic eruption, and flood with longer warning time, landslide is quite the opposite. The total time from the start of warning to actual landslide may be very short. The location of hazard zones may also be quite far and may be difficult to access. That is why enabling the community to perform independent evacuation led by the disaster preparedness team is important and needs to be supported. Fig. 6 shows the process of the evacuation drill. After the evacuation drill, in accordance with the seven sub-systems, local government signed a commitment memorandum containing the agreement to operate, sustain, and maintain the entire system.

**4 Discussions**

In the application of landslide early warning systems, the determination of hazard level and delivery of warning information is very crucial, as it determines the steps to be conducted by the disaster preparedness and response team and the local people. The application of appropriate SOP for command and communication is key to the whole management system (Calvello et al., 2016). The process of determining the hazard level and delivery of warning information can be explained simply through a series of mechanisms and decisions as follows. The system gives warning when monitoring devices detect landslide symptoms. It then goes through two paths until a final decision is reach to whether the evacuation should be carried out or not. The first path is information flow shown on the left side of Fig. 7. Field data logs created by monitoring devices are sent to the local server through a telemetry system (SMS, GSM, and radio frequency) to focal points (local leaders and trained key people). Trained local officers then conduct visual observation to each device and landslide-prone zones. The follow-up information is delivered to focal points, local authorities and the potentially impacted communities for "evacuation preparation". Generally monitoring the system can be conducted by different agencies/institutions, i.e. geological survey,

local government, research centers, and universities. Ideally, data from these various monitoring agencies is well integrated so that both national and local authorities can make an immediate decision on evacuation.

The second path is command flow (right side of Fig. 7), and it starts after information is received by the local authority. The local authority implements coordination with related stakeholders, e.g. local government, police/army, Red Cross, Search and Rescue (SAR) team, and emergency response unit. After the local authority officially declares the alert status, disaster preparedness teams then conduct SOP based on the status (Table 1). In WARNING status (Level 2) and EVACUATE (Level 3), results from monitoring devices can be directly conveyed by the local server to the community through sirens and signal warning lamps without local authority. The proposed methodology of information flow and command flow has been quite effective and strategic to improve the community resilience at the landslide vulnerable village. It is also crucial that the system should be developed through community participation utilizing the provision of both simple and low cost technology as well as real-time technology for early warning systems. This information flow and command system is a universal concept and adjustable to the conditions of each area or country.

Based on previous experiences when implementing this system since 2012, there are few key challenges and unexpected conditions that need to be prepared. As shown in Fig. 7 the factor of success in this particular landslide early warning systems is the multi-stakeholders participation. However, coordination among stakeholders can be very weak. In one implementation area, different early warning instruments were installed by various agencies with no integration. This caused the determination of alert level and evacuation decision is made based on partial monitoring data. One of the reasons this occurred was the variety of source of funding for the instruments. In addition, some of the activities conducted in a specific areas conventionally focused on data collection and research on landslide mechanism, instead of increasing the communities' capacity to respond to disaster. This is why landslide early warning systems standard became important as it included the implementation of proper coordination, among others, as part of the guidelines.

Another obstacle in general is the difficulties faced by disaster risk reduction programs. The level of local community awareness and preparedness is not constant at any given time. Usually after experiencing a disaster community preparedness levels can be high, however it is likely to decrease over time. The use and maintenance of the monitoring and warning devices tend to become neglected as community awareness decrease. It is therefore vital that communicate engagement is embedded throughout the development and implementation of a landslide early warning system. It is a great challenge for all stakeholders to develop the landslide early warning systems that not only last for short-term (1-2 years) but also to continue serve for its intended lifetime. The implementation of landslide early warning system will not stop the landslide occuring, but only to give warning to the local community. Hence it can be used as an entry point to increase the capacity of the local community. It is expected that through the implementation of this system, the local community knowledge can be increased and as such they will be able to independently conduct either structural or non-structural landslide disaster mitigation efforts.

## 5 Conclusions

Early warning systems are a vital part in disaster risk reduction. The main challenge for an early warning system is to implement it as a part of the community life. Therefore, in the landslide early warning systems, an integrated methodology to develop a universal standard for community-based early warning systems is proposed. This universal standard accommodates one of the priorities of the Sendai Framework described in the four elements of people-centered early warning systems, which is then developed into seven sub-systems of the landslide early warning system. The hybrid socio-technical approach is carried out to support the implementation of a landslide early warning system in Indonesia where the trial of this proposed methodology was done. Both approaches (technical and social), supported with continuing education and research, are expected to be able to involve all of the related stakeholders, reduce the cost of system implementation and

maintain its sustainability. The monitoring and warning service equipment that had been installed in various locations across Indonesia since 2012 is still in excellent condition until now by successfully implementing the newly proposed standards and maintenance methods.

It is important to know that landslides are common natural hazard triggers of disasters in remote areas the mitigation of which requires consideration of the technical, institutional, and socio-economic-cultural characteristics of the community. This proposed methodology is used to establish a common standard, starting with risk assessment and mapping, dissemination and communication, establishment of disaster preparedness and response teams, development of evacuation map, implementation of SOP, and installation of monitoring equipment. The standard is completed when the evacuation drill has been implemented and a commitment of the local authority and community on the operation and maintenance of an entire system is built. The standard emphasizes the joint role of central/local government and researchers/experts as facilitators to encourage the community to work independently on their preparedness and response capacities. Performing regular self-evacuation drills is important to maintain the community's spirit and alertness.

The primary issue that the adoption of this system addresses is that implementing the technical approach only is not effective to sustain disaster prevention. This failure often occurs when early warning system devices are installed by local authority/third party without local community involvement, so when the devices are triggered, the community lacks the ability to respond appropriately. The establishment and effective implementation of the seven sub-systems as a universal standard for landslide-prone countries would enhance current disaster risk reduction efforts. Also by increasing community involvement, the operation, maintenance, and sustainability of an entire disaster prevention system are secured early in the process.

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

| Close range monitoring instrument (in situ geotechnical instrumentation) | Data transfer by telemetry system: SMS, GSM, and Radio Frequency |
| Proximal and remote monitoring | |

**Figure 1: The newly proposed seven sub-systems for landslide early warning systems as the extraction of 4 key elements of people-centered EWS by UN-ISDR (2006).**

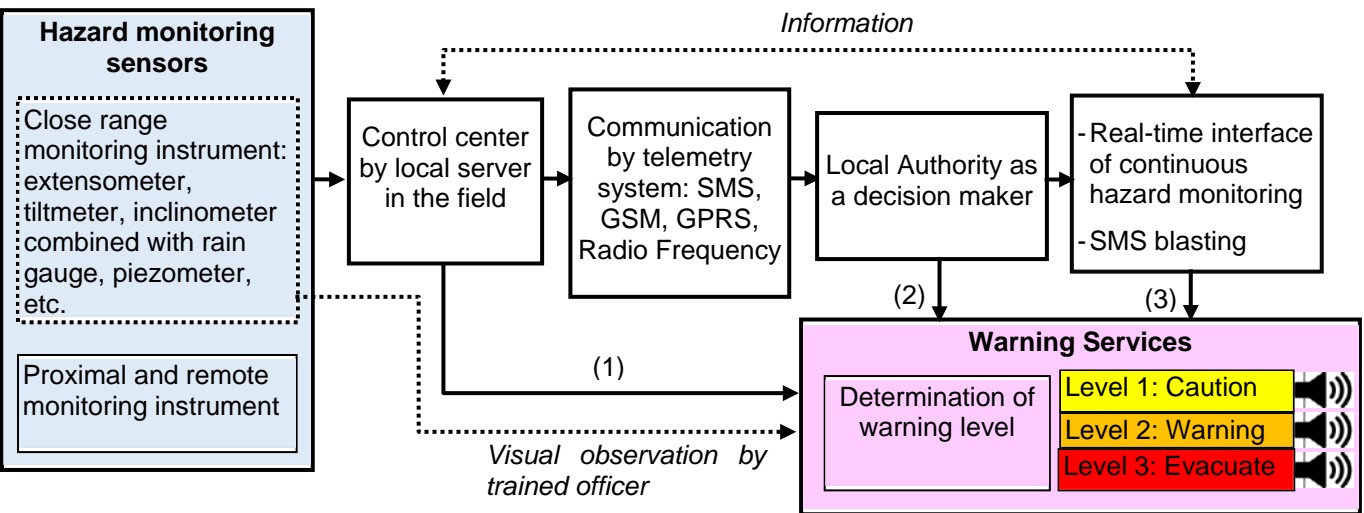

**Figure 2: Mechanism of data transmission among landslide monitoring and warning devices.**

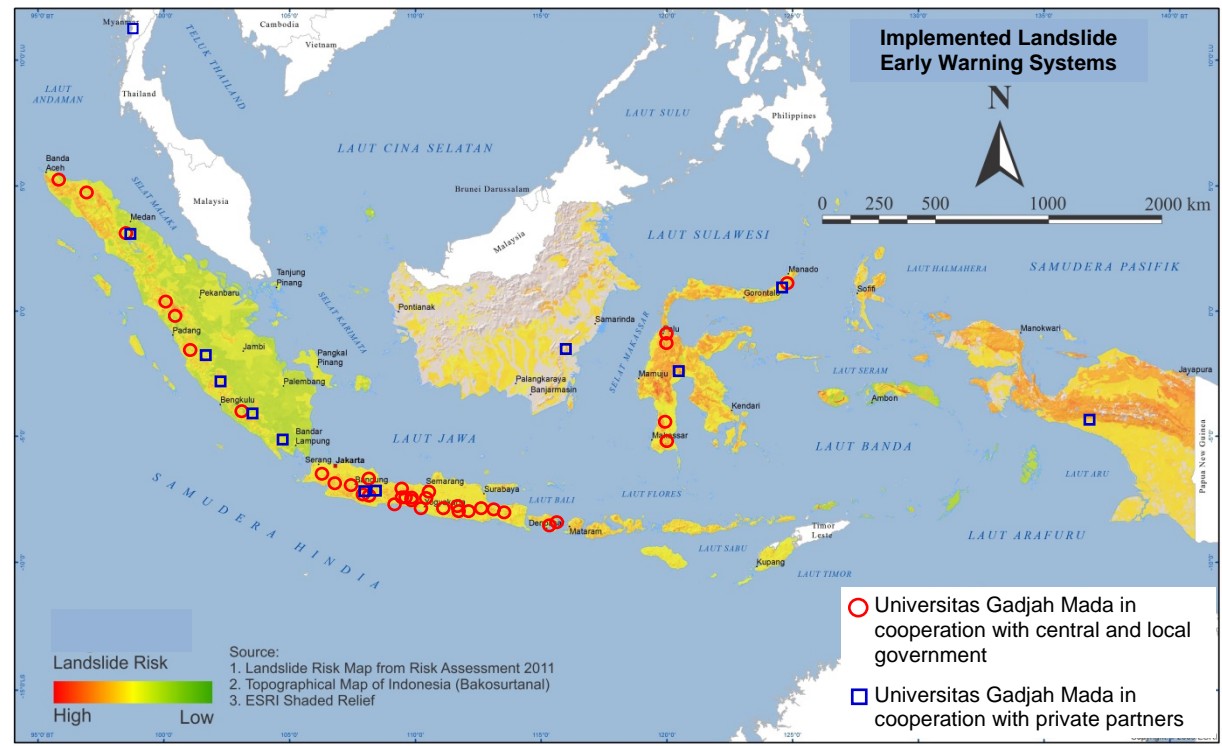

**Figure 3: The implementation of the new standard plotted on Indonesian Landslide Risk Map (BNPB, 2010).**

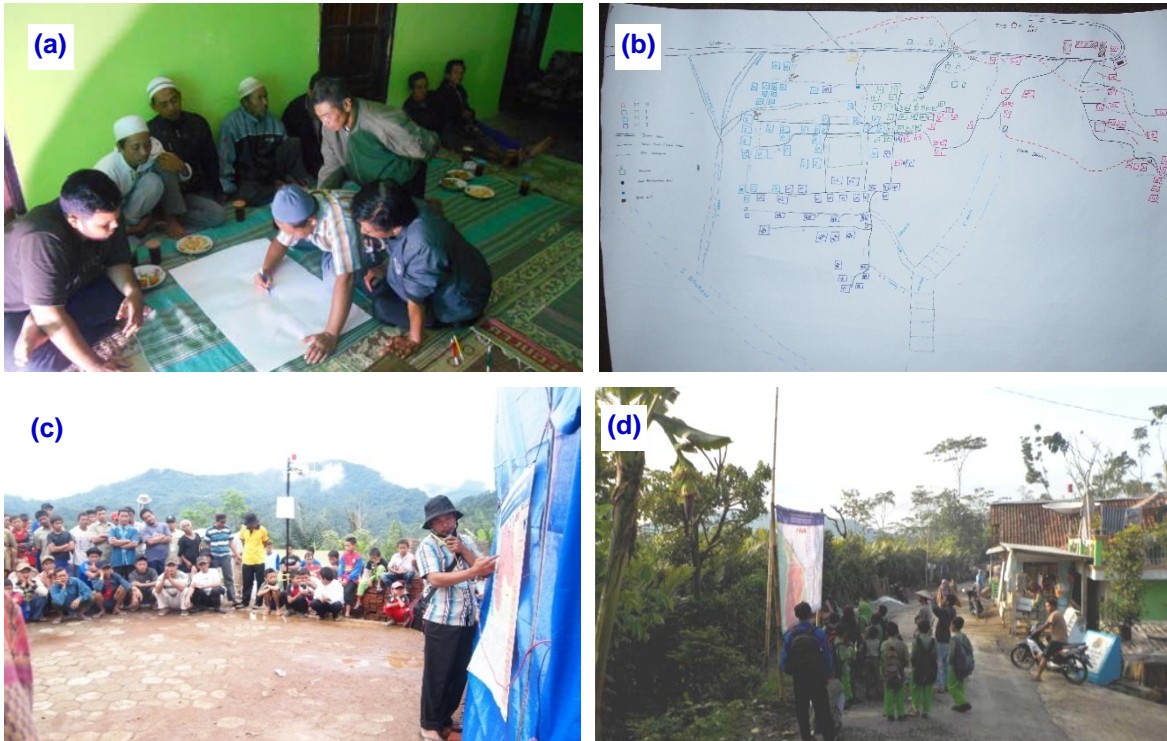

5 **Figure 4: Communication and dissemination process with local community: (a) The disaster preparedness and response team draw a community hazard map; (b) the newly developed community map; (c) team coordinator is explaining the map to the community; and (d) students read the evacuation map.**

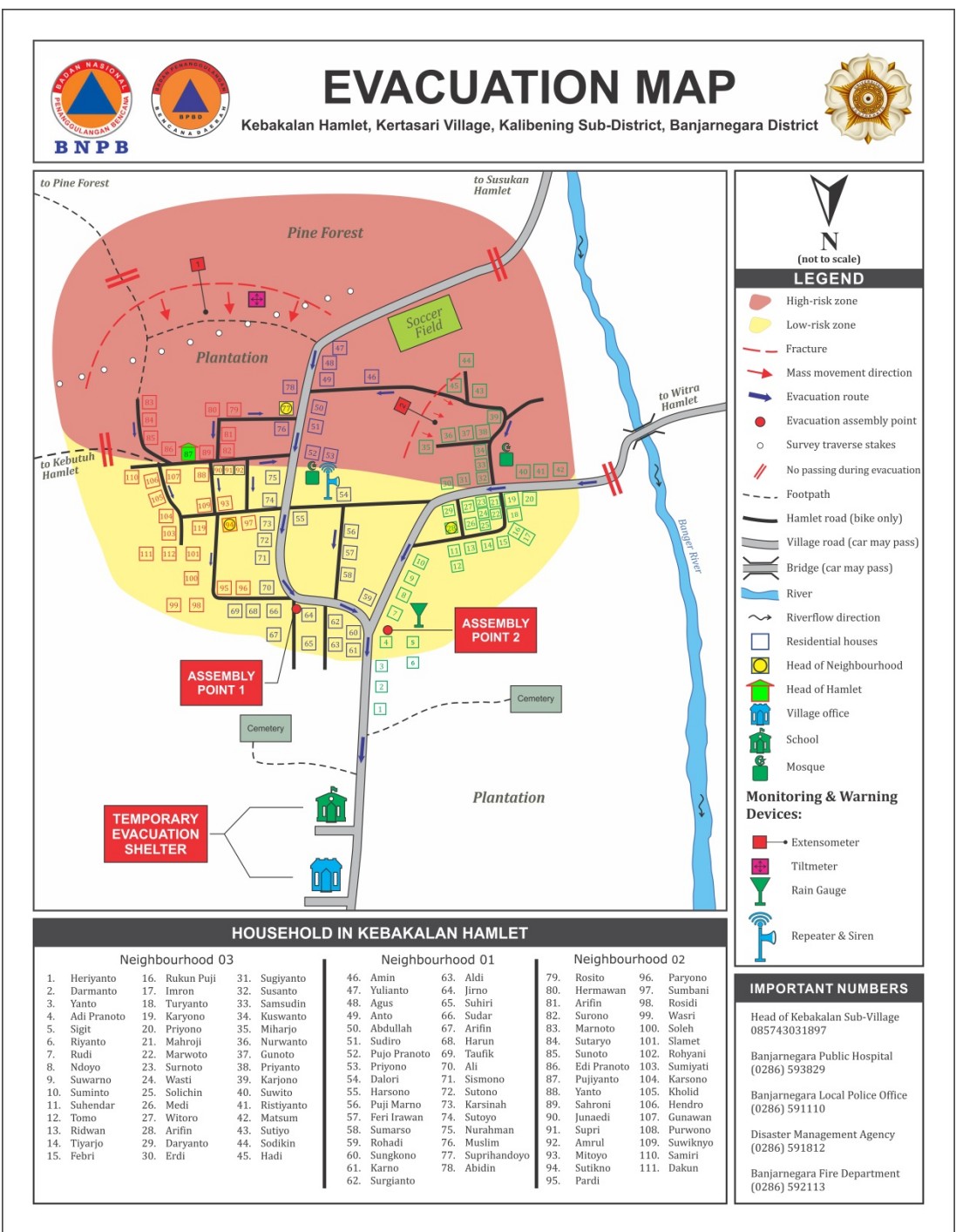

**Figure 5: Example of evacuation map.**

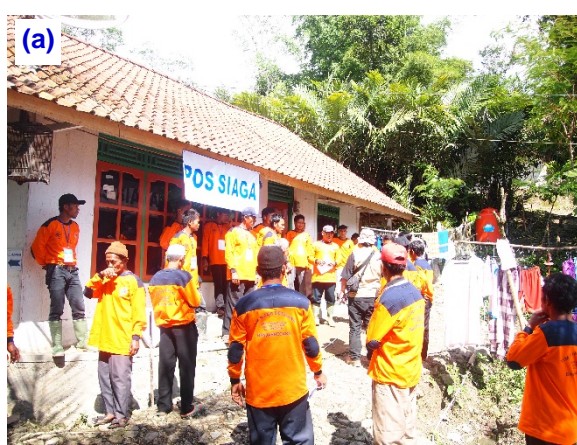 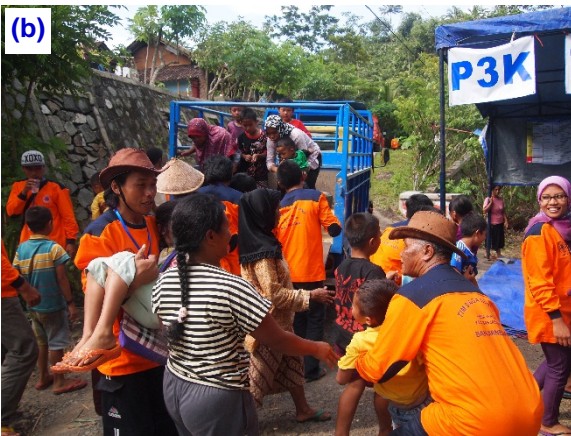

**Figure 6: Evacuation drill process: (a) coordination among the disaster preparedness and response team; (b) evacuation to the temporary shelter.**

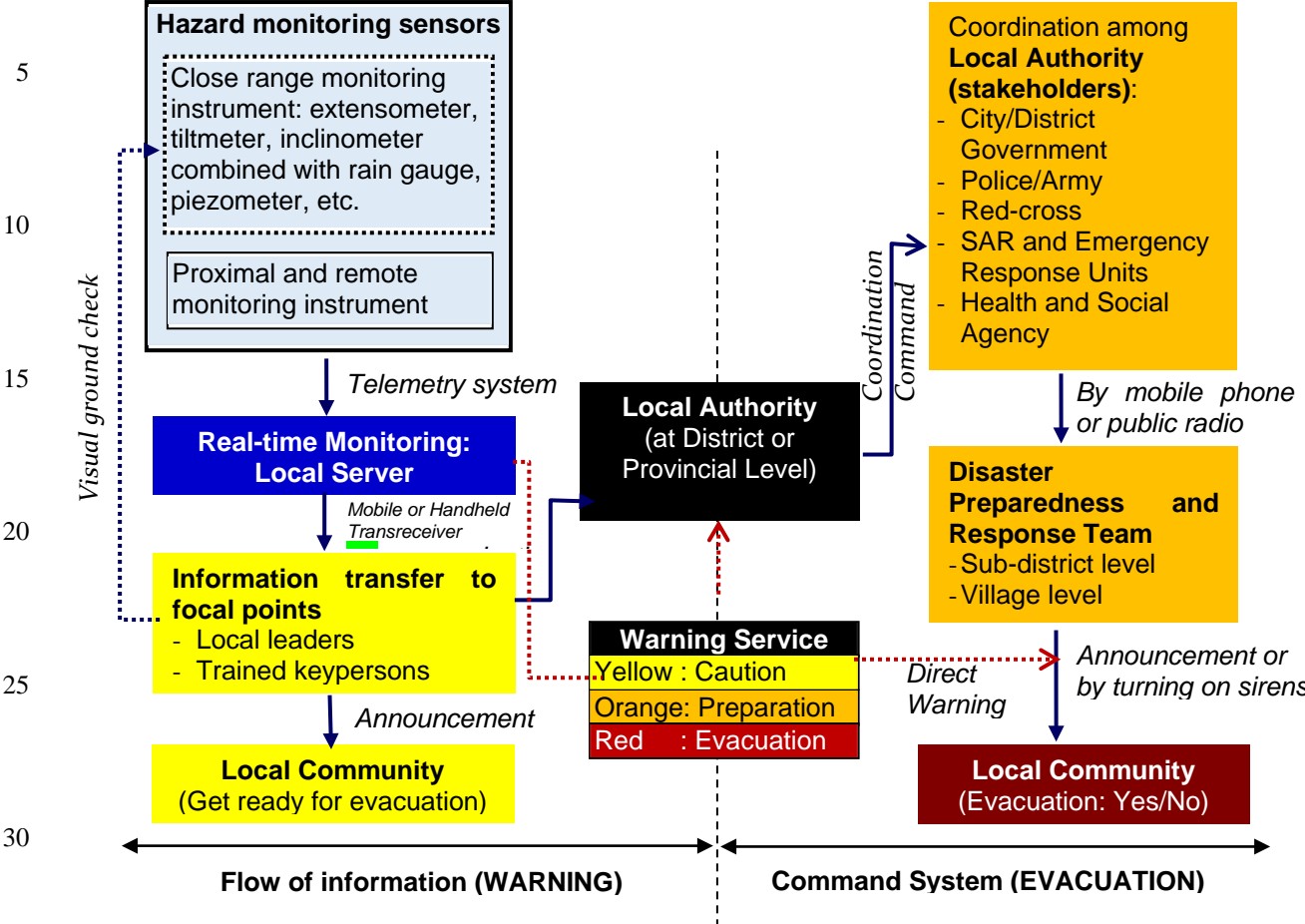

**Figure 7: Flow of information and command system for landslide early warning systems (after Fathani et al., 2014).**

**Table 1: Typical evacuation SOP to support a landslide early warning systems**

| Status/alert level | Criteria/ Sign | Action/response by the community | Action by the local authority |
|---|---|---|---|
| CAUTION: landslides possible (Level 1) | **Criteria**: determined by rainfall measurement (precipitation rate)<br><br>**Sign**: "blue" lamps and/or siren that sounds "caution, high rainfall" or other sound signs that show the lowest threat level or depending on the local conditions | • The team leader coordinates with the Disaster Preparedness Team.<br>• The data and information division checks the condition of the monitoring equipment and collects data of the community, and informs the alert level and encourages preparing essential items to bring.<br>• The Disaster Response Team provides periodic reports to the team leader. | • Receives report from the disaster preparedness team leader<br>• Checks the condition in the field and maintains coordination with the disaster preparedness team |
| WARNING: landslide likely (Level 2) | **Criteria**: determined by increased rainfall or groundwater, increased landslide indications in terms of ground surface or slip surface deformation<br><br>**Sign**: "orange" lamps and siren that sounds "warning, evacuate the vulnerable people" or other sound signs that show the increase of threat level to "warning" or depending on the local conditions | • The data and information division re-checks the condition of landslide and the monitoring equipment, and collects data of the community<br>• The team leader gives the vulnerable group an order to evacuate to the assembly point, with the help of the refugee mobilization division.<br>• The data section collects data of the vulnerable group in order to ensure that they have been evacuated.<br>• The security division is in charge of ensuring the security of the affected area. | • Receives report from the disaster preparedness team leader<br>• Checks the condition in the field and maintains coordination with the disaster preparedness team<br>• Provides support to the evacuated vulnerable group |
| EVACUATE: landslide occurrence imminent (Level 3) | **Criteria**: determined by increased rainfall or groundwater, increased landslide indications in terms of ground surface or slip surface deformation<br><br>**Sign**: "red" lamps and siren that sounds "evacuate" or other sound signs that show the highest threat level or depending on the local conditions | • The team leader gives all residents an order to evacuate to the assembly point, with the help of the refugee mobilization division.<br>• The data and information section checks the monitoring devices and collects data of the residents in the refugee camp. | • Receives report from the disaster preparedness team leader<br>• Checks the condition in the field and maintains coordination with the disaster preparedness team<br>• Provides emergency support to the evacuated residents |