# Peer review of "An integrated methodology to develop a standard for landslide early warning systems"

_Natural Hazards and Earth System Sciences, 2016_

## Referee Comment (RC1) · Anonymous Referee #1 · 21 Jun 2016

.:: General comments ::.

Dear authors,

I have been invited to review you manuscript titled "An integrated methodology to develop a standard for landslide early warning system" submitted to NHESS. Although I am terribly busy at the moment, have just finalised two other reviews and promised myself not to review anything in the next month, I could not resist as the topic of LEWS is very interesting and also my field of interest.

I very much appreciate the topic of the manuscript as it deals with landslide early warning systems but does not only focus on technical instruments. There are not many papers that highlight the importance of societal aspects of early warning and relate their works to all the four key elements of UN-ISDR.

[Figure]

Based on my review, I suggest some revisions. Please find my comments and suggestions below.

Best regards and keep up your good work, another reviewer

.:: Specific comments ::.

Language: Overall, the language and grammar are ok but there are still some issues that require a revision, preferably by a native speaker. I pointed out some mistakes but I guess that there are more. For example, in the title, it should be systems, not system.

Introduction: This section gives a reasonable introduction to the topic but then explains debris flow EWS as one example of an EWS. I find this strange because there are so many local and regional scale LEWS (not necessarily for debris flows) but 'the debris flow EWS' is noted as if it were the only EWS. You should rather give some examples of explicit LEWS examples. I suggest having a look at 'Thiebes 2012: Landslide Analysis and Early Warning Systems. Springer Thesis Series. DOI 10.1007/978-3-642-27526-5, download here: http://othes.univie.ac.at/15245/' and 'Thiebes, B. & Glade, T. (2016): Landslide early warning systems – fundamental concepts and innovative application. In: Aversa, S., Cascini, L., Picarelli, L. & Scavia, C. (eds.): Landslides and Engineered Slopes. Experience, Theory and Practice. Proceedings of the 12th International Symposium on Landslides, 1903–1911. DOI: 10.1201/b21520-238' – these are also a lot newer than the ones you mention later on (i.e. Glade and Wieczorek 2005 and Guzzetti 2008).

Scale: You do not mention in the manuscript that you are only dealing with local scale / single landslide LEWS. I think it would be good to point this out in the introduction.

Universality: You aim to develop a universal concept for LEWS implementations, however, you frequently refer to conditions that are typical for tropical conditions, e.g. the rainy season. So, I am asking myself how universal it is in the end. I actually think that the concept is also applicable to non-tropical countries and I therefore suggest to

slightly rewrite the sections where you refer to conditions that are related to tropical countries, e.g. by highlighting that in your case certain activities were carried out relative to the rainy season but that in other parts of the world the timing should be chosen differently.

Evacuation map: When I worked on LEWS in SE-Asia, I found a public evacuation map in the village centre to which everyone was invited to contribute by adding new landslide features that were found during field inspections. I am not recommending you to add this to the manuscript but rather to consider this for your upcoming activities. I guess that such a map is probably easier to accept by the local population.

Warning levels: The lowest level is already 'caution'. Why have you decided to not have a green level which means everyting is ok? Wouldn't that be useful? By having a green level, it is easy to see that the system is still working and it reminds people that they have a system. Again, nothing that needs to be integrated in the manuscript, rather meant for food for thoughts.

Legal aspects: Legal aspects are not dealt with - these frame the possibilities for an LEWS to a large extent and it should be mentioned that they are not discussed in the paper.

Figure 2: Would it make sense to include the inspection of a trained officer as a sensor? Presently, it seems like the oppinion of the trained officer is a shortcut to a warning, i.e. the decision on the warning is not issued by local authorities (decision maker). Similarily, it seems like the local control centre can directly issue a warning - another intentional shortcut? In any case, please elaborate the issueing of a warning a bit more and consider to add a couple of sentence on how a warning should be issued and what the chain of information and decision making is.

Installation of monitoring equipment (page9): I was a bit confused when I read this because for some time I thought that the sensors are only installed in the field when there is already a landslide warning. After reading it again, I think this is not the case –

you were only explaining that the SOP have to be implemented and then the sensors are installed. Maybe try to write this a bit clearer.

Warnings based on rain gauge measurements: I am not entirely happy with this as rainfall thresholds strongly depend on landslide types and previous knowledge. Without long time-series it is difficult to estimate the critical rainfall conditions. And even if you have long time series, what kind of threshold do you use? Hourly rainfall, daily rainfall, intensity-duration, antecedent rainfall... there are so many options. And then small landslides might be triggered by intense rainfall but large events rather because of rainy seasons with high total rainfall. How to determine the critical threshold if no monitoring system is yet available?

Determination of warning thresholds (not rainfall thresholds!) by experts: Should the local population or the mayor/head of the village be involved in this? This would increase the acceptance of false alarms and missed events. If the community wants to make sure that no alarms are missed, they have to deal with more false alarms; if they do not want false alarms, they might have some missed alarms for minor landslide events. Wouldn't the involvement of users increase the acceptance of false/missed alarms and the LEWS in general?

In your manuscript, you suggest 7 sub-systems for LEWS and relate this to the 4 elements of UN-ISDR. I completely agree with you that the 7 sub-systems make sense and include important activities. However, by adding them 'outside' of the 4 UN-ISDR elements you weaken the overall concept and are from my understanding not consequent. Actually, all the new-subsystems can comfortably be included within the 4 pillars of UN-ISDR. Sub-systems 3, 4 and 5, for example, are from my point of view part of the RESPONSE (UNISDR element 4). I recommend to redraw figure 1 and to add your sub-systems as parts of the 4 main elements. By doing so, your concept is better connected to the larger framework without losing any detail.

.:: Technical corrections ::.

[Figure]

Locations of the figures and the table are unclear as it is not mentioned in the text where they should appear.

P1 L8-9: landslides do not occur more often in areas of high population and low accessibility - it is rather that cause more disastrous effects there. Please rewrite a bit.

P2 L1-2: 'One example of efforts to implement the system is the debris flow early warning system.' - There are many examples of LEWS, not necessarily focussing on debris flows. And why 'the' debris flow EWS'? I suggest to change to 'one example are debris flow EWS'.

P2 L14: please add some more recent publications and dedicated LEWS reviews (I suggested some in the general comment section).

P4 L10: Capital letter in the beginning of the sentence.

P4 L27: the grammar in the sentence beginning here is not correct.

Citation Yueping. I think this Yueping Yin from the Geological Survey of China - where Yin is the last name. Please double check.

P6 L25: conventional and radar methods? This should be rather remote, proximal or close-range monitoring. Also correct in Figure 2.

P8 L2: end of the dry season. Do you have only troical countries such as Indonesia in mind or theoretically all parts of the world. Mentioning the dry season hints the former. Please check.

P8 L24: should be plural: surveys

P10 L6: use the abbreviation SOP again.

P11 L13: see SOP comment above.

Figure 3: There is a typo in the figure: 'Universitas Gadjah Mada ini cooperation with private sectors' should rather be 'Universitas Gadjah Mada in cooperation with private

[Figure]

partners/companies'. Are all the marked points implemented LEWS? This was not entirely clear to me. If yes, then please add this as a legend header, e.g. 'implemented landslide early warning systems' The word 'Legend' can be deleted as this is obviously a legend.

Figure 7: should be plural in the figure caption. And: mobile or HT - what does HT mean?

Table 1: should be plural in the caption. And: I would suggest to explain a bit what the different warning levels mean. For example: Caution = landslides possible; warning = landslide likely; evacuate = landslide occurrence imminent.

[Figure]

---

## Author Comment (AC1) · 26 Jun 2016

The authors would like to thank the Referee RC1 for his/her valuable comments. We believe this comment will be able to improve the scientific value of this manuscript. We will improve the writing of the manuscript according to the technical comments and will try hard to be able to address all the points in the specific comments as following:

Language: The authors agree and will improve the writing of the manuscript.

Introduction: The authors will improve the content of the Introduction section. We will refer to the references suggested by the referee and others regarding LEWS.

Scale: The authors will mention in the revision on this system is should be implemented for local scale/single landslide.

[Figure]

Universality: The authors will rewrite the sections regarding the conditions where this system will be applicable.

Evacuation map: Perhaps it is not clear in the manuscript but the evacuation map was made based on the result of discussion by the local people. The discussion process was led by a facilitator and need to comply with the basic guidelines as written in the manuscript and should be very simple and easily understood by the local people. We will add this information in the revision.

Warning levels: Number of warning levels is depending on the results of public consultation with the local community. In some pilot areas, the locals decide to have "green level" as the lowest level. Key activities during "green level" are (1) regular coordination between the disaster preparedness and response team and (2) regular check of the monitoring and warning devices. However, at most of pilot areas, the community decided to have "caution" as the lowest level. The authors will revise the manuscript to add more explanation about this "green level".

Legal aspects: The authors agree with the referee's comment. The only legal aspect in this system is that the importance of local government commitment to the implementation of LEWS in their region. We should clarify this point in the manuscript.

Figure 2: The trained officer role is to conduct a visual ground check on the monitoring equipment and warning device in order to identify if false warning happens (shown in dotted line). On the other hand, the trained officer might identify an obvious landslide movement in the field, but the equipment has a technical error to record the symptom. As shown in Fig. 2, there are three paths to issue the warning: (1) local control center; (2) local authority; (3) real-time interface by pushing the button by authorized officer. The author will revise the manuscript to elaborate the issuance of warning as being suggested by the referee.

Installation of monitoring equipment: Yes, we will clarify that in the revision of our manuscript.

Warning based on rain gauge measurement: Yes, it is a big challenge for the implementation of warning based rain gauge measurement. Some pilot areas used the thresholds suggested by previous researchers or the thresholds developed at other similar geological conditions. And we agree with the referee that there are so many options to determine a threshold. However this manuscript does not deal with the determination of the thresholds as warning criteria. In order to implement this system, the experts will have to decide which option will be used depending on the monitoring data availability. The manuscript will be revised to elaborate this issue.

Determination of warning threshold (not rainfall threshold?) by experts: Yes, we definitely agree with the referee. In fact, at all pilot areas, the disaster preparedness and response team has been involved in this. The acceptance of false/missed alarms is one of the key success of this system. The manuscript will be revised accordingly.

Figure 1: The authors agree with the suggestion by the referee. Figure 1 will be revised accordingly.

Technical corrections: The authors agree with the referee and will revise the manuscript based on the suggestions.

---

## Referee Comment (RC2) · K. Crowley (Referee) · 30 Jun 2016

This is an excellent contribution to the Nat Hazards Earth Syst. Sci Discussions. It is a thoughtful discussion on an interesting and important topic. I enjoyed reading this paper, in particular its interdisciplinary approach to addressing what is essentially an interdisciplinary issue that should cut across both physical and social realms. In particular I appreciate the elements that refer to the need for culturally sensitive and informed approaches, co-ownership/development and implementation of the system. These are all excellent points.

There are a few minor points that the authors may wish to reflect upon:

Page 1-4: The flow and grammar needs some further attention. It is always a challenge for researchers to write a paper that flows, and although these authors have done a

very good job, the first pages need further attention.

Page 2, paragraph 25: The glossary reference. This is an important point but the sentence somewhat downplays the importance by referring to a glossary – something common across many manuals. It would be good if the authors adjusted or added to this sentence to reflect the importance of having a common, clear and accepted technical and non-technical language within the standards – perhaps also add a point to why this is especially important in terms of landslides highlighting the differing applications of terminology and types.

Page 3, paragraph 20: Requires an additional line at the end of paragraph 19 to help the reader. For example, the following sections explore each of the 7 elements within the proposal EWS in turn.

Page 4, paragraph 15 Risk consciousness should be risk perception?

For the discussion/ case study:

It would be excellent to highlight some key challenges and unexpected observations from the case study. This I such an important element of the paper and it would be great to have more detail on how the EWS impact will be monitored going forwards. It is almost there but just needs a bit more depth.

Specially, it would strengthen the paper if the authors could reflect on the resource intensive nature of the EWS proposed. The authors have included a very important paragraph on government commitment (great!!) but it would be good to gain the authors reflection on which elements are resource intensive and for whom. The researches – both in terms of people and funding for equipment – seem to be a major issue in the implementation of EWS worldwide. How did the authors manage to overcome this and implement in so many locations? What were the challenges and what were some of the solutions?

I feel the discussion lacks a little depth at the moment – drawing on the authors reflections and 'top tips' for applications of the standard could provide some interest.

Otherwise, a very good paper on a very relevant topic.

---

## Author Comment (AC2) · 1 Jul 2016

The authors would like to thank Kate Crowley for her valuable comments. We believe this comment will be able to further improve the final manuscript. We will improve the flow and the grammar of the manuscript accordingly. The main feature in this paper is indeed its interdisciplinary approach (or "socio-technical approach" as the authors referred to in the manuscript). The authors will try hard to discuss the manuscript in more depth as the referee stated. The authors will also elaborate in the discussion about the challenges faced and solution implemented in the case study. Minor points in the referee comments will also be included in the revision of the manuscript.

---

## Author Response (AR1)

[revised manuscript text omitted]

**Comment [A7]:** Response to RC1 comment regarding listing "caution" as the lowest level.

**Comment [A8]:** To address comment from RC1 regarding the universality of LEWS implementation.

[revised manuscript text omitted]

Comment [A12]: Response to RC1 comment regarding the extent of the legal aspect in the manuscript.

[revised manuscript text omitted]

**Comment [A18]:** To address RC1 comment, the figure and its caption has been adjusted accordingly. HT stands for Handheld Transreceiver.

**Table 1**: Typical evacuation SOP to support a landslide early warning systems

| Status/alert level | Criteria/ Sign | Action/response by the community | Action by the local authority |
|---|---|---|---|
| CAUTION: landslides possible (Level 1) | **Criteria**: determined by rainfall measurement (precipitation rate)

**Sign**: "blue" lamps and/or siren that sounds "caution, high rainfall" or other sound signs that show the lowest threat level or depending on the local conditions | • The team leader coordinates with the Disaster Preparedness Team.
• The data and information division checks the condition of the monitoring equipment and collects data of the community, and informs the alert level and encourages preparing essential items to bring.
• The Disaster Response Team provides periodic reports to the team leader. | • Receives report from the disaster preparedness team leader
• Checks the condition in the field and maintains coordination with the disaster preparedness team |
| WARNING: landslide likely (Level 2) | **Criteria**: determined by increased rainfall or groundwater, increased landslide indications in terms of ground surface or slip surface deformation

**Sign**: "orange" lamps and siren that sounds "warning, evacuate the vulnerable people" or other sound signs that show the increase of threat level to "warning" or depending on the local conditions | • The data and information division re-checks the condition of landslide and the monitoring equipment, and collects data of the community
• The team leader gives the vulnerable group an order to evacuate to the assembly point, with the help of the refugee mobilization division.
• The data section collects data of the vulnerable group in order to ensure that they have been evacuated.
• The security division is in charge of ensuring the security of the affected area. | • Receives report from the disaster preparedness team leader
• Checks the condition in the field and maintains coordination with the disaster preparedness team
• Provides support to the evacuated vulnerable group |
| EVACUATE: landslide occurrence imminent (Level 3) | **Criteria**: determined by increased rainfall or groundwater, increased landslide indications in terms of ground surface or slip surface deformation

**Sign**: "red" lamps and siren that sounds "evacuate" or other sound signs that show the highest threat level or depending on the local conditions | • The team leader gives all residents an order to evacuate to the assembly point, with the help of the refugee mobilization division.
• The data and information section checks the monitoring devices and collects data of the residents in the refugee camp. | • Receives report from the disaster preparedness team leader
• Checks the condition in the field and maintains coordination with the disaster preparedness team
• Provides emergency support to the evacuated residents |

**Comment [A19]:** To address RC1 comment, the table has been adjusted accordingly.